# Honeybee Pollen Extracts Reduce Oxidative Stress and Steatosis in Hepatic Cells

**DOI:** 10.3390/molecules26010006

**Published:** 2020-12-22

**Authors:** Juan Esteban Oyarzún, Marcelo E. Andia, Sergio Uribe, Paula Núñez Pizarro, Gabriel Núñez, Gloria Montenegro, Raquel Bridi

**Affiliations:** 1Biomedical Imaging Center, School of Medicine, Pontificia Universidad Católica de Chile, ANID-Millennium Science Initiative Program-Millennium Nucleus for Cardiovascular Magnetic Resonance, Santiago 7820436, Chile; jeoyarzu@uc.cl (J.E.O.); meandia@uc.cl (M.E.A.); suribe@uc.cl (S.U.); 2Departamento de Farmacia, Facultad de Química, Pontificia Universidad Católica de Chile, Avda Vicuña Mackenna 4860, Santiago 7820436, Chile; pjnunez@uc.cl; 3Departamento de Ciencias Vegetales, Facultad de Agronomía e Ingeniería Forestal, Pontificia Universidad Católica de Chile, Avda Vicuña Mackenna 4860, Santiago 7820436, Chile; ginunez@uc.cl (G.N.); gmonten@uc.cl (G.M.)

**Keywords:** bee pollen, NAFLD, antioxidant, phenolic compounds, hepatoprotection

## Abstract

Nonalcoholic fatty liver disease (NAFLD) is a major cause of morbidity and mortality worldwide. Additional therapies using functional foods and dietary supplements have been investigated and used in clinical practice, showing them to be beneficial. Honeybee pollen from Chile has shown a large concentration of phenolic compounds and high antioxidant activity. In this work, we characterized twenty-eight bee pollen extracts from the central zone of Chile according to botanical origin, phenolic profile, quercetin concentration, and antioxidant activity (FRAP and ORAC-FL). Our results show a statistically significant positive correlation between total phenolic content and antioxidant capacity. Selected samples were evaluated on the ability to reverse the steatosis in an in vitro cell model using Hepa1-6 cells. The pollen extracts protected Hepa1-6 cells against oxidative damage triggered by 2,2′-azo-bis(2-amidinopropane) dihydrochloride (AAPH)derived free radicals. This effect can be credited to the ability of the phenolic compounds present in the extract to protect the liver cells from chemical-induced injury, which might be correlated to their free radical scavenging potential. Additionally, bee pollen extracts reduce lipid accumulation in a cellular model of steatosis. In summary, our results support the antioxidant, hepatoprotective, and anti-steatosis effect of bee pollen in an in vitro model.

## 1. Introduction

Nonalcoholic fatty liver disease (NAFLD) is a multifactorial disease, related to a complex living environment, heredity, and dietary habits. Patients with NAFLD have lower survival rates than healthy people of the same age and gender [1]. High-calorie diets and continued inactivity contribute to weight gain and promote the development of NAFLD [2]. Although the early stages of NAFLD may be reversible, studies have shown that 15% to 20% of patients with NAFLD can develop cirrhosis, and 30% to 40% of patients can suffer liver disease-related morbidity and mortality [3]. Weight loss through improved diet and increased physical activity has been the cornerstone therapy of NAFLD. Recent therapies using functional foods and dietary supplements have been shown to be beneficial [2,3]. There is evidence of a link between oxidative stress and the presence of NAFLD and its progression, particularly related to mechanisms such as mitochondrial dysfunction, endoplasmic reticulum (ER) stress, iron metabolism derangements, insulin resistance, and endothelial dysfunction [4,5,6].

The understanding of the molecular mechanisms responsible for lipid accumulation, oxidative balance impairment, and fibrosis in the liver could improve the therapeutic approach to decrease the risk of the disease progression. Antioxidant compounds, which modulate lipogenesis, inflammation, lipid oxidation and peroxidation, represent a new, attractive therapeutic approach for patients suffering from hepatic steatosis [5,7]. International recommendations for the treatment of nonalcoholic fatty liver suggest a reduction in the consumption of fats and sugars in the diet [5]. Additionally, the use of antioxidant micronutrients has been suggested as a potential treatment alternative, since oxidative stress is also pivotal in the progression of NALFD [7]. The Mediterranean diet, which is rich in antioxidant compounds such as polyphenols [8,9], has been shown to have a protective effect against this disease. Polyphenols are a heterogeneous group of compounds derived from plants. The hepatoprotective effects of some polyphenols have been reported, which modulate insulin resistance, oxidative stress, and inflammation [9]. These researchers suggest that the hepatoprotector mechanism may be related to the enhancement of the antioxidant capacity of cells [9,10].

Several groups have outlined the potential bioactive roles for honeybee pollen including antioxidant, immunomodulatory, cardioprotective, antimicrobial, and anti-inflammatory activities [11]. These activities are mainly attributed to phenolic compounds such as flavonoids. Poland bee pollen extract reduced and/or prevented the occurrence of steatosis and degenerative changes in the liver of mice caused by a high-fat diet, which may suggest a hepatoprotective role of bee pollen extract [12]. However, another study using *Schisandra chinensis* bee pollen on nonalcoholic fatty liver disease and gut microbiota in high-fat diet-induced obese mice suggested that phenolic compound present in the extract could attenuate the features of the metabolic syndrome [13].

In order to provide new and healthy sources of important food components, human society has developed so-called “functional food” which can be defined as a food prepared in order to afford different compounds (i.e., vitamins, fatty acids, proteins, carbohydrates, polyphenols, carotenoids, etc.) with the ability to have a positive influence on health [14]. The average content of polyphenols in pollen is around 1.6% (1.6 g/100 g) [15], which defines it as an excellent source of these bioactive compounds [14]. The botanical origin of pollen, its chemical composition, the amounts of major bioactive compounds such as polyphenols and carotenoids, and its antioxidant properties should be provided to better understand the impact of pollen addition in the formulation of functional food and feed products [14].

The central zone of Chile is one of five regions in the world that has a Mediterranean climate and is the largest producer of bee pollen in the country. The Valparaiso region (V region) is located in this area, which is covered by native vegetation characterized by a high level of endemism and biodiversity. Beehives are usually located in the “matorral” (the scrubs) communities, shrubby sclerophyllous vegetation that covers the slopes of the coastal range in the semiarid Mediterranean zone [16]. Our preliminary studies about chemical composition of bee pollen from the central zone of Chile showed the strong presence of phenolic compounds such as syringic acid, coumaric acid, myricetin, and quercetin [16].

The purpose of this study was to characterize according to botanical origin, phenolic profile, quercetin concentration, and antioxidant activity the honeybee pollen extracts from the central zone of Chile. From the characterized honeybee pollen extracts we selected six pollen samples with different concentrations of total phenols and we tested their ability to reverse steatosis in an in vitro cell model of liver steatosis.

## 2. Results and Discussion

### 2.1. Botanical Origin, Content of Phenolic Compounds, Flavonoid Compounds, Quercetin Concentration, ORAC-PGR, and FRAP Values of Honeybee Pollen

Table 1 shows the three predominant plant species in the samples, average values of total phenols (TP), flavonoids content (FC), concentration of quercetin (HPLC-DAD), and antioxidant activity (FRAP and ORAC-FL) in bee pollen extracts (BPEs) from the V region of Chile collected in 2018.

Botanical origin describes the presence of different plant sources used by bees to produce honeybee pollen. This description enables their classification as native/non-native/mixed and monofloral/bifloral/multifloral bee pollen, according to the Chilean norm (NCh 3255, 2011) [17]. The majority of samples analyzed corresponded to non-native multifloral followed by non-native monofloral and native monofloral. Among the non-native samples analyzed, the plant species *Brassica rapa* and *Eschscholzia californica* predominated. The predominant native species are *Cryptocarya alba* (peumo) and *Acacia caven* (espinillo). The complete information about botanical origin of the bee pollen samples performed by palynological analysis is shown in Appendix A.

In Table 1 the samples are listed in order of their TP concentration, from highest to lowest. Sample 1 presents the highest value for TP (1448 ± 116 GAE/100 g of bee pollen) while sample 28 presents the lowest value (102 ± 20 mg GAE/100 g of bee pollen). The content of flavonoids (FC) ranged between 504 ± 21 (sample 2) and 62 ± 9 mg QE/100 g (sample 28). The values of antioxidant capacity, evaluated by FRAP and ORAC-FL, were between 194 ± 7 (sample 1) and 19 ± 2 µmol TE/g bee pollen (sample 28), and between 492 ± 25 (sample 1) and 109 ± 49 µmol TE/g bee pollen (sample 22), respectively. The samples showed a considerable content of quercetin (HPLC-DAD) ranged from 409.94 ± 1.79 (sample 19) to 4.29 ± 0.45 mg/100 g fresh bee pollen (sample 26). 

The analysis of the 28 honeybee pollen extracts shows that there is a statistically significant positive correlation between total phenolic content and ORAC-FL (R^2^ = 0.41; *p* ≤ 0.03), and between the flavonoid content and FRAP (R^2^ = 0.40; *p* ≤ 0.03), and ORAC-FL (R^2^ = 0.50; *p* ≤ 0.01). Nonetheless, no correlation was observed between the quercetin (QE) concentration and antioxidant capacity (FRAP and ORAC-FL).

### 2.2. Phenolic Acids of Honeybee Pollen Extracts (BPEs) Determined by HPLC-DAD

From the results showed in the Table 1, six bee pollen samples were selected according to their content of phenolic compounds, to create three categories: high, medium, and low phenolic compounds content. Those samples were evaluated on the ability to reverse the steatosis in an in vitro cell model using Hepa1-6 cells and the concentrations of the most representative phenols (cinnamic acids, flavonols, flavone, and flavanone) were determined by HPLC-DAD; the results are depicted in Table 2. Fourteen phenolic compounds were quantified, and five compounds were found in all samples (syringic acid, cinnamic acid, rutin, myricetin, and quercetin). These data corroborated our previous studies that showed a considerable content of not only quercetin but also myricetin [16].

For selected samples, the analysis shows that there is a statistically significant positive correlation between flavonoid content and quercetin (R^2^ = 0.94; *p* ≤ 0.01) and myricetin concentration (R^2^ = 0.79; *p* ≤ 0.05). Previous studies proposed to use quercetin and myricetin as a quality indicator for honeybee pollen from this region of Chile [15], which can be used as quality control criteria for functional ingredients.

### 2.3. Cytotoxicity and Hepatoprotective Activity

In order to test the cytotoxicity of the six bee pollen samples selected, cultures of Hepa1-6 cells received treatment with ethanol extracts of bee pollen samples 1, 3, 4, 11, 22, and 28 at different concentrations (0.1, 1.0, 10 mg/mL). Besides these treatments, the experiment included a control group (untreated cells cultures), which was considered to present 100% cell viability. The viability of the cell cultures was determined as percentage of cell viability in relation to the control. The cytotoxicity was determined by Alamar blue of Hepa1-6 cells. The assays demonstrated that pollen extracts 4, 11, 22, and 28, at 0.1 mg/mL did not generate effects on cytotoxicity on these cells. At concentrations of 1 and 10 mg/mL, the cytotoxicity of the extracts increases (Figure 1). Accordingly, we used 0.1 mg/mL as a safe concentration for the hepatoprotective experiments.

The hepatoprotective effect of the extracts of bee pollen was evaluated under the same conditions as the cytotoxicity assay. The oxidation of Hepa1-6 cells was induced by the potent oxidant AAPH and this oxidation was performed in the presence and absence of 0.1 mg/mL of each tested pollen extract. The results showed a significant increase in cell viability in the presence of pollen extracts 1, 3, 4, 11, and 28 when compare with AAPH treatment alone, suggesting the protective effects of those bee pollen extracts against oxidative stress AAPH-induced free radicals’ accumulation in Hepa1-6 cells (Figure 2).

In order to test the potential effect of the pollen extracts in reversing steatosis, a classic steatosis cell model was used by loading Hepa1-6 cells with palmitic acid and oleic acid for 24 h [18,19]. When we loaded the hepa1-6 cells with lipids for 24 h they showed increases in the bodipy mark (Figure 3). Interestingly, when these cells were loaded with lipids and cotreated with different pollen extract, it was possible to observe a significant decrease in the accumulation of lipids in comparison with the control cells (Figure 3). 

It is possible to observe in the Figure 2 that the pollen extracts 1, 3, 4, 11, and 28 at a concentration 0.1 mg/mL showed a significant hepatoprotective effect against the oxidative damage caused by the free radical generator AAPH and these same pollen samples showed the best results in reducing the accumulation of lipids (Figure 3). We evaluated the correlation among the hepatoprotective effects of each pollen extracts and their total phenolic content, flavonoid content, quercetin concentration, and antioxidant capacity. The results indicated that there is a low positive correlation between steatosis reduction of total phenolic content (ρ = 0.26), flavonoid content (ρ = 0.23), and FRAP (ρ = 0.12). The antioxidant capacity (ORAC-FL) and quercetin concentration showed a moderate positive correlation with reduced steatosis, respectively ρ = 0.49 and 0.64 (Figure 4). The positive correlation between quercetin concentration and reduction of steatosis is a very interesting result since quercetin is present in all Chilean bee pollen samples studied in this and previous studies [16], corroborating the idea that this flavonoid is a good marker for determining the quality of Chilean honeybee pollen.

Antioxidants that can inhibit free radical generation are important in terms of protecting the liver from chemical-induced damage by stabilizing the antioxidant systems in the cell. In vivo and in vitro studies have demonstrated the promising preventative and therapeutic effects of plant phenolics in a range of liver diseases, particularly in NAFLD [10]. Flavonoids are able to control de novo lipogenesis, inhibiting lipogenic proteins and increasing lipolytic proteins, and they are also effective scavengers of reactive species that are elevated in pathological states and metabolic disorders such as NAFLD. Recent studies have shown the beneficial effects of quercetin. In HepG2 cells and in animal models, an increase in mitochondrial biogenesis was observed [5]. Epicatechin and apigenin have been reported to protect the liver from NAFLD, which are associated with their effects on insulin resistance and for signaling the way to anti-inflammation as well as antioxidant action [20,21]. Dihydromyricetin supplementation improves glucose and lipid metabolism as well as various biochemical parameters in patients with nonalcoholic fatty liver disease, and the therapeutic effects of dihydromyricetin are likely attributable to improved insulin resistance and decreases in the serum levels of tumor necrosis factor-alpha, cytokeratin-18, and fibroblast growth factor 21 [20,22]. Furthermore, myricetin, a similar flavonoid that is in a higher concentration in our samples, between 35 and 775 mg/100 g pollen, can efficiently relieve hepatocyte necrosis, inflammation and oxidative stress, and in vitro studies have suggested that remarkably attenuated H_2_O_2_-triggered hepatotoxicity and ROS generation [23].

In conclusion, bee pollen extracts showed hepatoprotection against a known free radical generator such as AAPH, suggesting that the antioxidant potential of pollen is protecting against this insult. On the other hand, bee pollen extracts reduce lipid accumulation in a cellular model of steatosis. Those positive effects were positive correlated with the pollen’s quercetin concentration. Our results highlight the potential use of bee pollen as a hepatoprotective and anti-steatosis treatment in fatty liver diseases and could be a potential micronutrient to prevent and eventually reverse NAFLD, which affects a large proportion of Western countries’ populations. There is a worldwide prevalence of 25.24%, with a higher percentage in the Middle East and South America, and a lower prevalence in Africa [24].

## 3. Materials and Methods

### 3.1. Honeybee Pollen Samples

Twenty-eight samples from the V region of Chile (GPS coordinate 33°3′47″ S, 71°38′22″ W) were provided as vacuum packed when fresh and frozen at −20 °C by associated beekeepers. The samples were collected during the dry seasons of 2018. The determination of botanical origin was performed using the palynological analysis method described in Chilean Regulation (NCh3255, 2011) [16]. To determine botanical origin, specific literature [25,26] and the botanical bee pollen catalog at the Laboratory of Botany (Department of Plant Sciences, Faculty of Agronomy and Forest Engineering, Pontificia Universidad Católica de Chile, Santiago, Chile) were consulted.

### 3.2. Extract Preparation and Phenolic Characterization

The honeybee pollen extracts (BPEs) was obtained by ultrasonic extraction (Elmasonic S 10 H ELMA) bath at 37 kHz frequency and 240 W using one gram of fresh honeybee pollen in analytical grade absolute ethanol (10 mL) at room temperature (25 °C) for 10 min [27]. The mixture was centrifuged at 3130× *g* for 5 min and filtered using Whatman No. 1 paper. This procedure was repeated three times for each sample, and the collected extracts were combined to a final volume of 50 mL (1 g/50 mL). BPEs were stored at −80 °C in the dark until use.

The total polyphenols content in the extracts was measured using the Folin Ciocalteu’s method. The gallic acid equivalents (GAE) were used to express the content of total polyphenols, i.e., milligrams per 100 g of bee pollen (mg GAE/100 g) [16]. The flavonoid content measured by AlCl_3_ method was calculated as milligrams of quercetin equivalents (QE) per 100 g of bee pollen (mg QE/100 g) [16].

### 3.3. Antioxidant Capacity Assay

The ferric reducing antioxidant power (FRAP) of the pollen extracts was determined as previously described by Bridi et al. [16]. The working FRAP solution was prepared daily by mixing 10 parts of acetate buffer (0.3 M pH 3.6), 1 part of 10 mM TPTZ (2,4,6-tri(2-pyridyl-s-triazine, Sigma), and 1 part of 20 mM ferric chloride. Aliquots of 270 µL FRAP solution were mixed with 30 µL of diluted BPE (1:100). The reaction mixtures were incubated for 30 min at 37 °C and the absorbance was measured at 594 nm using a Cytation 5 multi-mode microplate reader from BioTek Instruments, Inc. (Winooski, VT, USA). As positive controls, ethanol solutions and Trolox (5–30 µM) were used. The results are expressed as µmol Trolox equivalents per g of bee pollen (µmol TE/g). Values are reported as mean ± SD of 3 independent determinations.

The antioxidant capacity of BPEs was measured by using the ORAC-fluorescein (ORAC-FL) assay conducted on the basis of a report by Ou et al. [28] and adapted to the use of a fluorescent microplate reader (Cytation™ 5 from BioTek Instruments Inc.) [16]. The fluorescein consumption was assessed by the decrease in fluorescence intensity of the sample (excitation: 493 nm; emission 515 nm). AAPH was used as the peroxyl generator and µM Trolox as a standard (2–10 µM). The results are expressed as µmol Trolox equivalents per 100 g of bee pollen (µmol TE/100 g). Values are reported as mean ± SD of 3 independent determinations.

### 3.4. HPLC-DAD Analysis

BPE samples were analyzed with a Hitachi Chromaster 5000 series high-performance liquid chromatography (HPLC) instrument equipped with an autosampler and a photodiode array detector (DAD) (Hitachi, Tokyo, Japan). The HPLC system was controlled by Chromaster system manager V1.2. The BPEs were separated using a mobile phase mixture of (A) methanol, (B) acetonitrile, and (C) 0.1% aqueous formic acid. The composition of the mobile phase mixture varied by employing the following HPLC stepwise gradient program: 0–10 min 20% B, 80% C; 10.1–40 min 7.5% A, 25% B 67.5% C; 40.1–50 min 15% A, 25% B, 60% C; 50.1–65 min 15% A, 45% B 40% C, and returned to starting conditions during the following 15 min. The column used was a 250 mm × 4.6 mm i.d., Purospher STAR RP-18 endcapped with a guard column of the same type while the flow rate was kept at 0.8 mL/min and oven column set at 35 °C. The absorbance of 10 µL eluate was monitored with a DAD detector set in the 210–550 nm range and the chromatograms were integrated for all standards and BPE samples at 290 nm. Quantification was performed with calibration curves using commercially available standards (range 5–250 µM of each component). All analysis was performed in triplicate for standards and BPE samples [16].

### 3.5. Cytotoxicity and Hepatoprotective Activity In Vitro

Cell line: Hepa1-6 cells were used [29,30]. These cells were grown in 75 cm^2^ flasks using DMEM with high glucose content, supplemented with 10% FBS (fetal bovine serum) and 1% antibiotic solution. The cells were kept in a humidified atmosphere with 5% CO_2_–95% air at 37 °C. 

Cytotoxicity and hepatoprotective activity: Hepa1-6 cells were seeded at a density of 50,000 cells per well in 96-well plates. After 24 h, cells were incubated with ethanol and ethanol extracts of pollen (1, 3, 4, 11, 22, and 28) at different concentrations (0.1, 1, 10 mg/mL). The induction of cell damage was carried out using 2,2′-Azobis (2-amidinopropane) dihydrochloride (AAPH, 0.2 mM). Triton X100 1% was applied as a positive control. Cell mortality was determined by reduction of resazurin (Alamar blue assay) and measuring fluorescence (560 nm excitation/590 nm emission) using a Cytation 5 multi-mode microplate reader from BioTek Instruments Inc. (Winooski, VT, USA) [31]. The results are expressed as a percentage of the control conditions of three independent experiments and four replicates per experiment.

### 3.6. Statistical Analysis

All data represent the mean values ± SD of at least 3 independent experiments, each conducted in triplicate. Pearson and Spearman correlation and analyses were carried out using Origin Pro 8 software (MA, US). Statistical comparisons between groups were evaluated with one-way ANOVA followed by Tukey post-hoc test. In the in vitro experiment, the Mann-Whitney statistical test was performed. A value of *p* < 0.05 was considered to be significant.

## Figures and Tables

**Figure 1 molecules-26-00006-f001:**
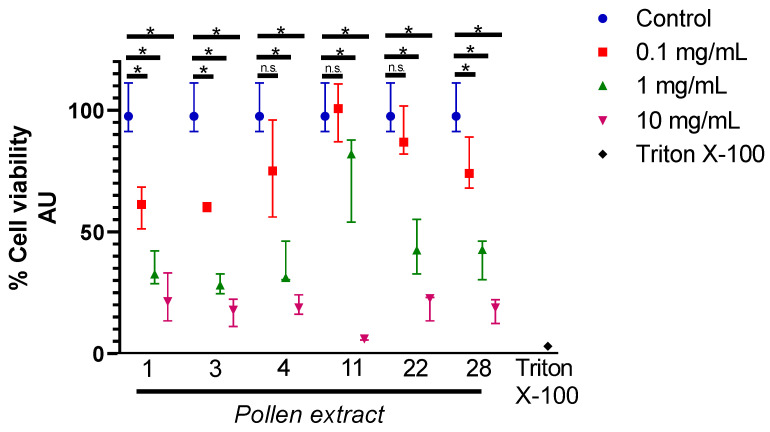
Maximum concentration of pollen extracts. Cell viability evaluated by Alamar blue of Hepa1-6 cells treated with ethanol extracts of pollen 1, 3, 4, 11, 22, and 28 at different concentration (0.1, 1, 10 mg/mL). Negative control, cells treated with TritonX-100 at 1% for 10 min. Arbitrary Units (AU). Cell viability expressed as a percentage of viable cells relative to the control cells (untreated cells). Data are shown as median ± first quartile and third quartile (*n* = 3). Mann-Whitney statistical test was performed between the control group and different pollen concentrations. The asterisks (*) represent statistically significant differences (*p* < 0.05). n.s.: not significant.

**Figure 2 molecules-26-00006-f002:**
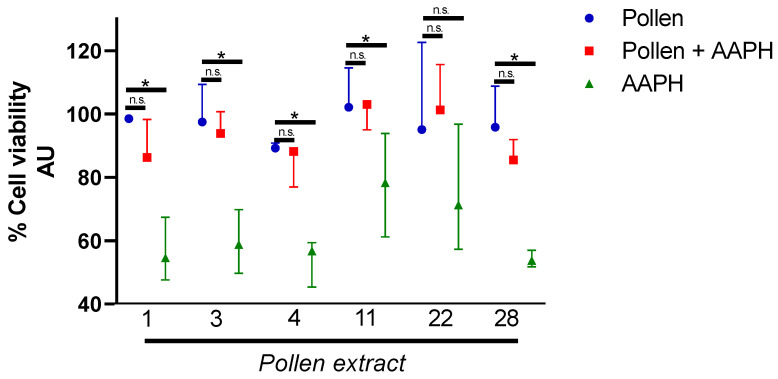
Pollen extract prevents 2,2′-Azobis(2-amidinopropane) dihydrochloride (AAPH) induced cell death. Cell viability evaluated by Alamar blue of Hepa1-6 cells treated with ethanol extracts of pollen 1, 3, 4, 11, 22, and 28 at a concentration of 0.1 mg/mL and co-treatment with AAPH for 24 h at 0.2 mm. Cell viability expressed as a percentage of viable cells relative to cells treated only with pollen extract. Data are shown as median ± first quartile and third quartile (*n* = 3). Mann-Whitney statistical test was performed between the cells treated with pollen, pollen + AAPH, and AAPH. The asterisks (*) represent statistically significant differences (*p* < 0.05). n.s.: not significant.

**Figure 3 molecules-26-00006-f003:**
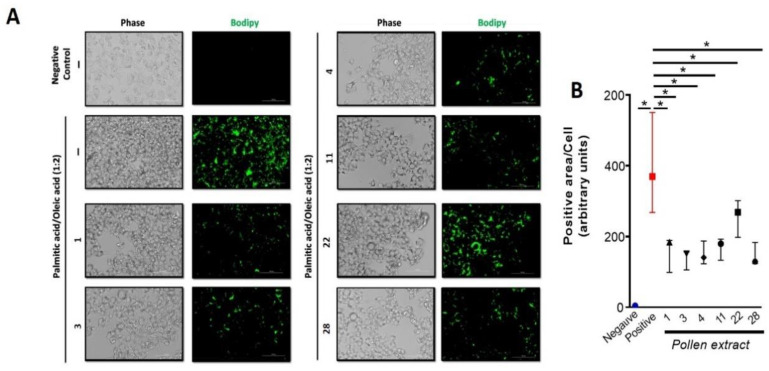
Pollen extracts reduce steatosis in an in vitro cell model. (**A**) Bodipy lipid staining in Hepa1-6 cells line treated with ethanol extracts of pollen at a concentration of 0.1 mg/mL 1, 3, 4, 11, 22, and 28. The cells were pre-treated with pollen extracts for 2 h, prior to loading with lipids (palmitic/oleic acid) for an additional 24 h. (**B**) The quantifications are in arbitrary units to estimate the bodipy lipid staining in Hepa1-6 line treated with ethanol extracts of pollen. Data are shown as median ± first quartile and third quartile (*n* = 3). Mann-Whitney statistical test was performed between the positive control and different pollen extract. The asterisks (*) represent statistically significant differences (*p* < 0.05).

**Figure 4 molecules-26-00006-f004:**
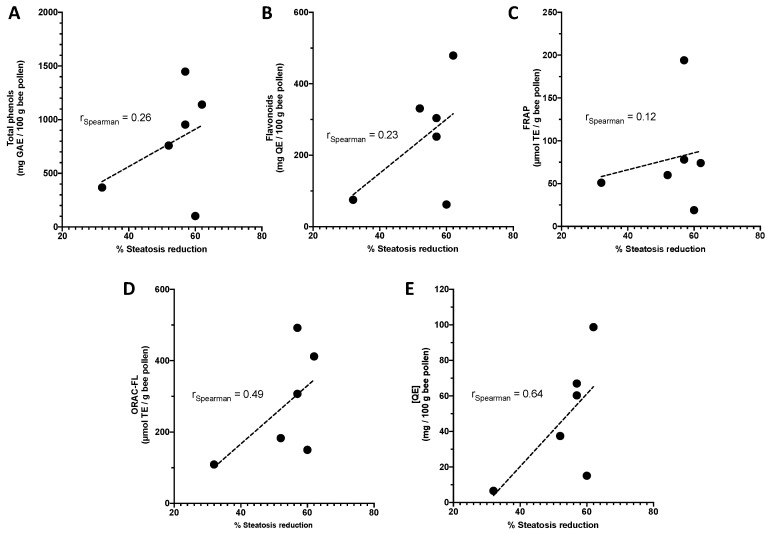
Correlation coefficient (ρ) from Spearman correlation analysis between hepatoprotective effects and (**A**) total phenols content, (**B**) flavonoid content, (**C**) antioxidant capacity (FRAP), (**D**) antioxidant capacity (ORAC-FL), and (**E**) quercetin concentration [QE].

**Table 1 molecules-26-00006-t001:** Average values of total phenols (TP), flavonoids, FRAP, ORAC-FL, and quercetin (QE) concentration in bee pollen extracts from the V region of Chile collected in 2018.

Samples	Predominant Plant Species			Total Phenols	Flavonoids	FRAP	ORAC-FL	[QE]
	First	Second	Third	(mg GAE/100 g Bee Pollen)	(mg QE/100 g Bee Pollen)	(µmol TE/g Bee Pollen)	(µmol TE/g Bee Pollen)	(mg/100 g Bee Pollen)
1 *	*Eschscholzia californica*	*Adesmia arborea*	*Taraxacum officinale*	1448 ± 116	252 ± 18	194 ± 7	492 ± 25	60.32 ± 0.34
2	*Eschscholzia californica*	*Brassica rapa*	*Eucalyptus* sp.	1301 ± 12	504 ± 21	111 ± 4	312 ± 22	99.37 ± 0.18
3 *	*Brassica rapa*	*Cryptocarya alba*	*Sophora macrocarpa*	1141 ± 30	479 ± 6	74 ± 1	412 ± 89	98.70 ± 18.45
4 *	*Brassica rapa*	*Cryptocarya alba*	*Anthemis cotula*	954 ± 83	304 ± 8	78 ± 15	307 ± 61	66.99 ± 2.82
5	*Brassica rapa*	*Adesmia arborea*	*Colliguaja odorifera*	920 ± 38	261 ± 14	101 ± 1	224 ± 27	16.04 ± 0.08
6	*Acacia caven*	*Eschscholzia californica*	*Acacia caven*	848 ± 2	273 ± 12	70 ± 4	253 ± 8	84.22 ± 20.77
7	*Brassica rapa*	*Cryptocarya alba*	*Azara* sp.	8379 ± 55	346 ± 8	76 ± 15	398 ± 34	65.13 ± 0.76
8	*Brassica rapa*	*Cryptocarya alba*	*Azara* sp.	780 ± 17	326 ± 20	91 ± 16	397 ± 40	39.49 ± 0.75
9	*Brassica rapa*	*Colliguaja odorifera*	*Amomyrtus luma*	767 ± 22	168 ± 10	86 ± 4	371 ± 36	133.68 ± 1.10
10	*Brassica rapa*	*Eschscholzia californica*	*Cryptocarya alba*	767 ± 109	205 ± 7	107 ± 6	350 ± 38	4.55 ± 0.52
11 *	*Azara celastrina*	*Mix*	*Cryptocarya alba*	759 ± 7	331 ± 26	60 ± 2	183 ± 11	37.44 ± 0.54
12	*Cryptocarya alba*	*Brassica rapa*	*Eschscholzia californica*	758 ± 72	251 ± 12	84 ± 8	282 ± 68	25.05 ± 0.33
13	*Brassica rapa*	*Colliguaja odorifera*	*Eschscholzia californica*	744 ± 15	172 ± 9	96 ± 9	338 ± 27	175.22 ± 0.96
14	*Brassica rapa*	*Adesmia arborea*	*Cryptocarya alba*	725 ± 44	337 ± 14	66 ± 3	197 ± 35	296.08 ± 3.43
15	*Brassica rapa*	*Eucalyptus* sp.	*Raphanus sativus*	717 ± 15	149 ± 16	98 ± 1	371 ± 45	115.74 ± 0.99
16	*Brassica rapa*	*Cryptocarya alba*	*Hypochaeris radicata*	715 ± 52	186 ± 14	80 ± 17	281 ± 87	162.38 ± 0.51
17	*Brassica rapa*	*Cryptocarya alba*	*Schinus latifolius*	588 ± 12	150 ± 8	66 ± 13	203 ± 39	121.87 ± 0.74
18	*Eschscholzia californica*	*Brassica rapa*	*Schinus latifolius*	580 ± 5	165 ± 17	56 ± 3	231 ± 47	145.98 ± 1.18
19	*Cryptocarya alba*	*Brassica rapa*	*Eschscholzia californica*	540 ± 21	88 ± 8	61 ± 4	277 ± 99	409.94 ± 1.79
20	*Eschscholzia californica*	*Cryptocarya alba*	*Anthemis cotula*	536 ± 10	260 ± 6	35 ± 9	277 ± 129	22.34 ± 0.1
21	*Brassica rapa*	*Lythrum hyssopifolia*	*Robinia pseudoacacia*	518 ± 51	102 ± 5	54 ± 7	185 ± 74	63.19 ± 0.36
22 *	*Brassica rapa*	*Eschscholzia californica*	*Cactaceae*	368 ± 19	75 ± 7	51 ± 2	109 ± 49	6.59 ± 0.19
23	*Cryptocarya alba*	*Eschscholzia californica*	*Taraxacum officinale*	364 ± 14	176 ± 18	38 ± 6	166 ± 45	11.2 ± 0.14
24	*Brassica rapa*	*Schinus latifolius*	*Azara celastina*	350 ± 41	122 ± 4	41 ± 3	158 ± 40	5.24 ± 0.45
25	*Brassica rapa*	*Cryptocarya alba*	*Azara* sp.	344 ± 27	128 ± 6	39 ± 10	191 ± 28	20.61 ± 0.36
26	*Cryptocarya alba*	*Eschscholzia californica*	*Malva* sp.	315 ± 15	156 ± 1	27 ± 2	216 ± 44	4.29 ± 0.45
27	*Cryptocarya alba*	*Eschscholzia californica*	*Anthemis cotula*	286 ± 34	153 ± 3	26 ± 1	242 ± 42	6.7 ± 0.18
28 *	*Cryptocarya alba*	*Anthemis cotula*	*Baccharis linearis*	102 ± 20	62 ± 9	19 ± 2	150 ± 67	15.08 ± 0.3

* Select samples to carry out studies on the ability to reverse the steatosis and chemical characterization.

**Table 2 molecules-26-00006-t002:** Phenolic acids and flavonoids of honeybee pollen extracts (BPE) determined by HPLC-DAD ^1^.

**Samples**	**mg Phenolic Acids/100 g Pollen**						
	**Chlorogenic**	**Caffeic**	**Syringic**	**Coumaric**	**Sinapic**	**Ferulic**	**Cinnamic**
1	0	0	6.62 ± 0.05	0.8 ± 0.12	0	6.76 ± 0.57	20.18 ± 0.1
3	0	0	1.77 ± 0.1	0.94 ± 0.02	0	15.34 ± 0.36	16.14 ± 0.21
4	0	0.93 ± 0.02	1.35 ± 0.02	0.41 ± 0.23	0.98 ± 0.42	6.38 ± 0.33	21.42 ± 0.31
11	15.92 ± 0.18	0	13.64 ± 0.13	0	16.7 ± 0.3	20.2 ± 0.29	10.15 ± 0.04
22	26.21 ± 0.37	5.67 ± 0.08	5.81 ± 0.2	0.2 ± 0.01	5.36 ± 0.22	1.03 ± 0.51	2.81 ± 0.91
28	16.76 ± 1.16	0	2.21 ± 0.4	0	19.05 ± 5.78	0	5.42 ± 0.23
**Samples**	**mg Flavonoids/ 100 g Pollen**						
	**Epicatechin**	**Rutin**	**Myricetin**	**Quercetin**	**Apigenin**	**Rhamnetin**	**Catechin**
1	7.95 ± 0.1	13.98 ± 0.07	775.89 ± 1.87	60.32 ± 0.34	34.62 ± 2.71	0	0
3	0	7.19 ± 0.18	185.65 ± 1.43	98.7 ± 18.45	767.24 ± 83.92	0	0
4	38.42 ± 0.14	36.2 ± 0.68	121.41 ± 3.31	66.99 ± 2.82	0	0	0
11	21.4 ± 0.05	74.25 ± 2.93	96.46 ± 0.41	37.44 ± 0.54	21.05 ± 2.56	13.35 ± 2.88	15.8 ± 0.34
22	0	2.74 ± 1.43	28.51 ± 0.95	6.59 ± 0.19	13.09 ± 0.06	0	0
28	0	17.57 ± 0.4	35.49 ± 3.08	15.08 ± 0.3	6.98 ± 0.63	80.37 ± 3.19	17.96 ± 0.44

^1^ Data are expressed as mg/100 g fresh bee pollen and the values represent the means ± SD (*n* = 3).

## Data Availability

Samples of the compounds are not available from the authors.

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
