# Peer review of "Honeybee Pollen Extracts Reduce Oxidative Stress and Steatosis in Hepatic Cells"

_molecules, 2020, doi:10.3390/molecules26010006_

Round 1

Reviewer 1 Report

REF Manuscript ID: molecules-1027958

Honeybee pollen extracts reduce oxidative stress and steatosis in hepatic cells

The paper shows a multilevel study, from honeybee pollen  botanical origin, through  average concentration  of total phenols (TP), flavonoids, FRAP, ORAC-FL, and quercetin (QE) to evaluation of beepollen samples to the ability to reverse the steatosis in an in vitro  cell model using Hepa1-6 cells

As a promising results of the study, the authors found that at a concentration of 0.1mg / mL of bee pollen can have a significant hepatoprotective effect against the oxidative damage caused by the free radicals and in reduction of  the accumulation of lipids. 

The authors indicate that supplementation with honeybee pollen can  improve glucose and lipid metabolism as well as various biochemical parameters in patients with nonalcoholic fatty liver disease

This a good designed study with properly given hypotheses. The methods are appreciate as well as statistical analyses. 

I found only several small mistakes that needs correction

78 - area, which is cover by a native … - correct for -  is covered by a native…….

85 honeybee pollen extracts … correct for - the honeybee pollen extracts……..

120 - honey bee pollen. Use homogenous orthography (spelling)  throughout the entire text

Fig 1. The indicators for concentrations are not clear. It is difficult to distinguish between the concentration levels

I recommend to move to main body text the Table S2.  Correlation coefficient (r) from Pearson correlation analysis…….and Table S3.  Correlation coefficient (r) from Spearman correlation analysis…… Moreover, Table S3 should be named as Figure, not the Table. However, it is an editorial comment.    

Reviewer 2 Report

  1. How you chose extraction conditions for polyphenols from pollen? Can you specify the value of ultrasonic amplitude used for extraction?
  2. The contents in myricetin of the analyzed samples are much higher than in quercitin. However, based on previous specialized reports, you have chosen quercitin to calculate correlation coefficients. Did you tried to see if the correlation coefficients for myricetin are not better given its high concentration?
  3. Table 1, samples 19 total polyphenols 540 ± 21 mg GAE and total quercetin 409.94 ± 1.79mg QE. Can you check the results?
  4. Sample 28, total polyphenols from table 1,  102 ± 20mg GAE,  total flavonoids 62 ± 9 mg QE, Total phenolic acids from table 2 approx.  43mg, flavonoids approx. 173 mg? Can you check the results?
  5. In my opinion, figure 1 is not clear enough. Can you make any adjustments?
